# Improvement of Temperature Distribution Uniformity of Ready-to-Eat Rice during Microwave Reheating via Optimizing Packaging Structure

**DOI:** 10.3390/foods12152938

**Published:** 2023-08-02

**Authors:** Chai Liu, Liuyang Shen, Huiran Liu, Xue Gong, Chenghai Liu, Xianzhe Zheng, Shuo Zhang, Chen Yang

**Affiliations:** 1College of Engineering, Northeast Agricultural University, Harbin 150030, China; liuchai@neau.edu.cn (C.L.); feiyanghero@163.com (L.S.); 13796517111@163.com (H.L.); liuchenghai@neau.edu.cn (C.L.); zsmax1020@163.com (S.Z.); ycmax00@163.com (C.Y.); 2College of Light Industry, Harbin University of Commerce, Harbin 150028, China; kahnannie@163.com

**Keywords:** microwave reheating, ready-to-eat rice, temperature uniformity, metalized packaging

## Abstract

The taste quality of ready-to-eat rice is influenced by the uniformity of temperature distribution during microwave reheating. The temperature distribution uniformity of ready-to-eat rice loaded in a rectangular lunch box is investigated under microwave reheating. The results show that with a 10–80 °C temperature increase in the ready-to-eat rice, the thermal conductivity increases, dielectric constant, and specific heat increase and then decrease, while the dielectric loss factor decreases and then slightly increases. The microwave-heating process of ready-to-eat rice exhibits a clear ‘corner effect’, and the observed ‘hot spot’ results in poor temperature uniformity in ready-to-eat rice. A metalized packaging structure design is subsequently proposed to ameliorate the temperature non-uniformity. According to comparative results of four metalized packaging forms, the spray film volume and film thickness corresponding to film volume are developed as $3.5\times 10^{-4}$ mL/${mm}^{2}$, 0.30 mm, respectively, which levels off the difference in temperature to improve the temperature distribution uniformity of ready-to-eat rice by microwave reheating.

## 1. Introduction

Microwave heating technology provides a high heating rate, energy efficiency, and a bactericidal effect [1]. It is widely used in the processing of agricultural products in grain drying [2], the pretreatment of fruits, and vegetables [3,4,5], and is also suitable for reheating prepared foods (such as chicken dishes, cooked rice, and ready meals) [6,7,8]. Due to the characteristics of microwave heating and the complexity of food composition [9,10], microwave reheating may cause issues such as uneven temperature distribution and unpredictable quality changes [11].

The dielectric properties of food materials absorb microwave energy, where violent movement of the internal polar molecules subjected to the microwave field results in sharp friction and collision among molecules. This process consequently generates volumetric heat and elevations in temperature [12,13]. Previous studies have reported that the shape of the food can affect temperature uniformity distribution during microwave heating [14,15]. For example, microwave refraction and reflection in rectangular food, as well as the microwave focusing effect at the corner of food, cause the temperature to rise faster at the corner [16]. Due to microwave refraction and focusing effects, spherical food (such as eggs, potatoes, and tomatoes) experiences central overheating when heated in a microwave [17]. The excessive absorption of microwave energy may lead to the ‘thermal runaway’ phenomenon [18]. An intermittent microwave drying process was introduced to effectively solve the ‘thermal runaway’ problem in mushrooms caused by a sharp increase in microwave volume heat in the late drying period [19]. The electric field distribution inside the microwave cavity also influences microwave heating uniformity, where the standing wave mode formed by the spatial transmission of an electric field causes a ‘hot spot’ to appear in a localized area of the material [20,21]. Improving the heating uniformity of microwave-reheated food has significant practical implications for food microwave processing.

The uniformity of the interior temperature distribution of food is related to the overheating or uneven heating phenomenon under microwave reheating, which results in poor quality of food processing [22,23]. As a result, investigating the effect of food composition and structure on the law of microwave energy absorption and temperature distribution is useful for improving temperature distribution uniformity in microwave reheating and product quality. Ready-to-eat rice (RER), which has a high porosity and low homogeneity, is frequently employed as a staple food in fast food. The investigation of the temperature distribution of RER during microwave reheating is helpful to provide a parameter basis for improving the microwave reheating effect of fast food.

Food absorbs microwave energy and then transforms into heat energy during the microwave reheating process. Heat energy raises the temperature of food and facilitates water diffusion and evaporation. Changes in food properties affect the temperature distribution during the reheating process, and dielectric properties determine food’s ability to absorb and transform microwave energy [22]. Studying the change of food dielectric properties is helpful to analyze the influence of microwave energy absorption on temperature distribution. In the process of microwave reheating, changes in thermal properties can indicate the law of heat consumption and transport. Food characteristic indexes are useful for clarifying the heat and mass transfer mechanism of food during microwave reheating, analyzing the causes of uneven temperature distribution, and providing a reasonable basis for improving uneven temperature distribution. Furthermore, active packaging has the ability to change the features of the electric field distribution in the RER under microwave heating. Consequently, this makes it possible to build a realistic active packaging structure to improve the uniformity of microwave heating of RER [5]. Microwave food packaging, as an efficient and low-cost means of improving microwave heating uniformity, offers vast application potential in the microwave food business.

Therefore, the main objective of this study is to design metalized packaging structures that ameliorate the non-uniformity of temperature. The research objectives are developed as follows:(1)To obtain the variation of parameters, such as moisture content, density, porosity, thermal characteristics, and dielectric properties, under various changes in temperature.(2)To analyze the temperature distribution of RER during microwave reheating, and identify the causes of uneven temperature distribution.(3)To design a metalized packaging structure that ameliorates the temperature non-uniformity.

## 2. Materials and Methods

### 2.1. Materials

Fresh rice was provided by Harbin Commercial Doctoral Science and Technology Development Co., Ltd. (Harbin, China). Polypropylene (PP) rectangular boxes with suitable temperatures from −20 to 120 °C were purchased from the website, the volumes of the boxes were 500, 650, 750, and 1000 mL, with bottom sizes 138 × 86 mm^2^, and heights 40, 44, 50, and 65 mm, respectively.

### 2.2. Preparation and Measurement of Materials

#### 2.2.1. RER Preparation

RER samples were prepared according to GB/T 15682-2008 [24]. Fresh rice (No. 2 fresh rice, Dr. Shang Inc., Harbin, China) was stored in a 4 °C refrigerator for 24 h before being washed three times, soaked for 30 min at a rice-to-water ratio of 1:1.3 (g/g), and cooked with an electric cooker (SR-CW15, Panasonic Inc., Osaka, Japan). PP rectangular boxes (500, 650, 750, and 1000 mL) were filled with cooked rice at a ratio of 0.83 ± 0.03 g/mL. The boxed rice’s surface was kept smooth, sealed and cooled to room temperature, and then stored for 24 h at 4 °C.

#### 2.2.2. Moisture Content Measurement

The moisture content of RER was calculated using the 105 °C oven drying method [25]. The oven temperature was set to 105 °C and preheated for 0.5 h. After 1 h of drying in the oven, the box lid was covered and placed in a dry dish for 0.5 h, and the weight of the aluminum box was weighed. Filling the aluminum box with 2–5 g RER, drying it in the oven for 1 h, then covering it and putting it in a drying dish for 0.5 h before weighing; repeating the drying and weighing process until the mass difference between the two times does not exceed 2 mg, which is constant weight; each group of experiment was repeated three times.

#### 2.2.3. Density Measurement

The apparent density of RER was calculated by the volume discharging method [26]. Weighing 2–5 g of the RER and putting it in a measuring cylinder filled with 100 mL of ethanol solution (20%), the volume change equals the RER’s volume.

#### 2.2.4. Porosity Measurement

Porosity is defined as the ratio of pore volume in porous materials to the total volume in their natural state [27]. The porosity of RER was determined by Equation (1).
(1)$$P=\frac{V_{0}-V}{V_{0}}\times 100\%=\left(1-\frac{\rho }{\rho_{0}}\right)\times 100\%$$
where $V_{0}$ is the apparent molar volume of RER in its natural state (m^3^), *V* is the volume in the absolute density of RER (m^3^), $\rho_{0}$ represents the apparent density of RER (kg/m^3^) and ρ represents the absolute density of RER (kg/m^3^).

#### 2.2.5. Thermal Characteristics Measurement

A thermal characteristic analyzer (KD2 Pro, Decagon Inc., Pullman, WA, USA) was introduced to measure the thermal conductivity and specific heat of RER over a temperature ranging from 10 to 80 °C, with a measurement interval of 10 °C [28], and the measurement was performed three times. Following cold storage, the RER was placed in a water bath pot that achieved the desired temperature, and the center of RER was measured in real-time with a handheld infrared thermometer (FLUKE 563, Fluke Inc., Everett, WA, USA). When the center temperature matched the predetermined temperature, the thermal characteristic analyzer’s sensor (SH-1, Fluke Inc., USA) was employed to measure the RER’s thermal conductivity K value and specific heat C value of the material.

#### 2.2.6. Dielectric Properties Measurement

A network vector analyzer (E5071C ENA, Agilent Inc., Santa Clara, CA, USA) was used to measure the dielectric properties of RER using the coaxial measuring method with an underwater bath temperature ranging from 10 to 80 °C with a measurement interval of 10 °C and the measurement was repeated three times [29]. Following smashed rice powder at a mesh size of 30, the fresh rice was soaked for 30 min with a rice-to-water ratio of 1:1.3 (g/g), stirred once every 10 min, and then cooked. The cooked rice was cooled at room temperature then placed in a 100 mL beaker, wrapped with plastic wrap, and refrigerated for 24 h. The refrigerated RER was placed in a temperature-controlled water bath, and the temperature in the core of RER was monitored in real-time with a handheld infrared thermometer (FLUKE 563, Fluke Inc., Everett, WA, USA). When the RER’s center temperature matched the predetermined temperature, the dielectric properties were measured with the network vector analyzer.

#### 2.2.7. Microwave Penetration Depth Calculation

The microwave penetration depth is the position at which the energy decays to 1/e when the materials are irradiated with microwaves [29], as calculated by Equation (2).
(2)$$d_{p}=\frac{C_{0}}{2\pi f\sqrt{2\varepsilon^{\prime }[\sqrt{1+{\left(\frac{\varepsilon \prime \prime }{\varepsilon^{\prime }}\right)}^{2}}-1]}}$$
where $C_{0}$ is the free-space light velocity in m/s, $\varepsilon^{\prime }$ represents the relative dielectric constant of RER and $\varepsilon^{\prime \prime }$ is relative dielectric loss of RER.

#### 2.2.8. Measurement Position Setting

Rectangular RER was divided into three layers A_1_, B_1_, and C_1_, and 13 representative points were chosen for each layer, as shown in Figure 1. Considering the axisymmetric rectangle in each layer, 13 points were equivalent to 45 measurement points at each layer to represent the entire layer of data.

#### 2.2.9. Measurement Method of Temperature Data

As illustrated in Figure 2, microwave reheating began after placing the RER (with volumes of 500, 650, 750, and 1000 mL, respectively) into the microwave workstation (the frequency is 2.45 GHz) and setting the microwave power (700, 800, 900, and 1000 W) and heating time (30, 60, 90, 120, 150, and 180 s). The real-time temperature of key points is acquired at 0.6 s intervals using an optical fiber temperature sensor (FOT-L-SD-C1-F1-M2-R1-ST, FLIR Inc., Tigard, OR, USA) and FISO software (MWS, FLIR Inc., USA). The temperature distribution of layers A_1_, B_1_, and C_1_ were measured using an infrared thermal imager (FLIR E95, FLIR Inc., Tigard, OR, USA) at 30, 60, 90, 120, 150, and 180 s of reheating time, respectively.

#### 2.2.10. Microwave Power Selection

Setting the lunch box volume at 650 mL, the microwave power to 700, 800, 900, and 1000 W, and the quality of the RER to 540 ± 20 g, the appropriate microwave power was determined according to temperature evaluation.

#### 2.2.11. Volume of Lunch Box Selection

Setting the microwave power of 800 W, the lunch box volumes of 500, 650, 750, and 1000 mL, reheating time of 180 s, and filling the lunch box with RER at the ratio of 0.83 ± 0.03 g/mL, the appropriate volume was selected based on temperature evaluation.

#### 2.2.12. Temperature Evaluation

The microwave reheating effect was evaluated by the temperature change of 13 key points during reheating and the temperature distribution (average temperature, temperature uniformity, maximum and minimum temperature) of the three layers undergoing reheating. Equation (3) was used to calculate the layer average temperature of RER, and Equation (4) was used to calculate the holistic average temperature of RER.
(3)$${\bar{T}}_{1}=\frac{1}{N}\sum_{m=1}^{N}T_{m}$$
(4)$${\bar{T}}_{2}=\frac{1}{n\times N}\sum_{i=1}^{n}\sum_{j=1}^{N}T_{ij}$$
where *N* is the number of key points in a layer, *n* is the layer number, $T_{m}$ and $T_{ij}$ is temperature of each key point in a layer, respectively.

Temperature distribution uniformity can be estimated using the difference between the maximum and minimum temperature, where a smaller temperature difference indicates greater temperature distribution uniformity [30]. The temperature difference method ignored the influence of temperature in different regions of the RER on temperature distribution uniformity. To evaluate uniformity while ignoring the influence of rising mean temperature, the temperature uniformity coefficient (*COV*) was introduced to define the ratio of temperature standard deviation to rising mean temperature. The rising mean temperature was the difference between the mean temperature and the initial temperature [31]. Equation (5) was introduced to calculate the layer *COV*_1_ of RER, and Equation (6) was used to calculate the holistic average *COV*_2_ of RER. The smaller the *COV* means the higher the uniform temperature distribution.
(5)$$COV_{1}=\frac{\sqrt{\frac{1}{N}\sum_{i=1}^{N}\left(T_{i}-\bar{T}\right)}}{\bar{T}-T_{0}}$$
(6)$$COV_{2}=\frac{\sqrt{\frac{1}{n\times N}\sum_{i=1}^{n}\sum_{j=1}^{N}\left(T_{ij}-\bar{T}\right)}}{\bar{T}-T_{0}}$$
where *N* is the number of key points in a layer, *n* is the layer number, $T_{ij}$ represents the key points’ temperature in the layer (°C), $T_{0}$ is the initial temperature value (°C), and $\bar{T}$ represents the key points’ average temperature (°C).

#### 2.2.13. Metalized Packaging Preparation

Six g CMC was fully stirred and dissolved in 100 mL deionized water before being heated in a 50 °C water bath for 3 h. After adding 12 g Al powder and then stirring, the solution was placed in the 50 °C water bath for another 0.5 h to produce an aluminum film solution. The aluminum film solution was sprayed on the lunch box’s appropriate structure, and the metalized packaging box was prepared by drying in an oven at 80 °C and then standing at room temperature.

### 2.3. Data Processing and Analysis

The software Origin (2018, OriginLab Inc., Northampton, MA, USA) was used to create data graphs, and SPSS Statistics (V21.0, IBM Inc., Armonk, NY, USA) was employed for statistical analysis. The software SigmaPlot (V12.5, Systat Inc., San Jose, CA, USA) was used to establish temperature distribution images of RER along the *x*-, *y*-, and *z*-axes. A three-dimensional effect visual map was created in Photoshop (CC 2018, Adobe Inc., San Jose, CA, USA) by placing temperature distribution images from the same direction on the same coordinate axis as shown in Figure 3.

## 3. Results and Analysis

### 3.1. Density of RER

The moisture content change in A_1_, B_1_, and C_1_ layers (Figure 1) under microwave reheating are shown in Table 1. Variance analysis suggested that the influences of microwave reheating duration and sampling position on RER’s moisture content were insignificant (*p* > 0.05). During cold storage, RER will air shaft and harden, resulting in retrogradation, whereas microwave reheating had the potential to modify the phenomenon [32]. The average moisture content of RER was measured as 63.23 ± 0.87%. Due to the dependency of density on moisture content, the apparent density of RER was 880 ± 3.29 kg/m^3^, absolute density was 1670 ± 6.42 kg/m^3^, and the porosity as calculated by Equation (3) was 47.31 ± 0.56%.

### 3.2. Thermal Characteristics of RER

As shown in Figure 4a, the thermal conductivity of RER increases significantly at temperatures ranging from 10 to 80 (*p* < 0.05). The increase in temperature enhanced moisture diffusion in RER to produce thermal expansion, which reduced porosity, and increased density, thus reducing contact thermal resistance between RER particles and increasing the thermal conductivity [33].

The specific heat of RER tends to increase and then decrease with increasing temperature in the range of 10–80 °C (Figure 4b) (*p* < 0.05). The absorption of microwave energy in RER was converted into heat energy during initial heating, which raised the temperature and accelerated the diffusion of internal moisture. Rising temperature caused an increase in specific heat of materials. In the later period of the heating process, the absorption of heat energy in the RER was primarily consumed via moisture evaporation, the temperature change was no longer significant (*p* > 0.05), and the specific heat decreased gradually and tended to be stable [21].

### 3.3. Dielectric Properties of RER

The dielectric constant of RER exhibits a significant difference in the 10–80 °C range when measured at 2.45 GHz (Figure 5a) (*p* < 0.05). The directional polarization rate of polar molecules in RER increased at temperatures between 10–30 °C, and as a result, the dielectric constant increased with temperature [34]. When the temperature was above 30 °C, the dielectric constant decreased with temperature, thus the directional polarization rate has a negative relationship with temperature.

The dielectric loss of RER gradually decreases and then becomes temperature stable in the range of 10 °C to 80 °C at 2.45 GHz (Figure 5b) (*p* < 0.05). Under microwave heating, the vibration frequency of polar molecules in RER (such as water and starch) lags behind the microwave frequency of 2.45 GHz [34]. The lag degree of vibration frequency increased with temperature, which is significantly lower than the critical vibration frequency, to impel dielectric loss increase with temperature.

The microwave penetration depth inside RER tends to increase and then decrease with temperature between 10–80 °C at 2.45 GHz (Figure 5c), with a significant difference (*p* < 0.05), which was affected by the dielectric properties of materials related to the dielectric constant and dielectric loss. The retrogradation of RER gradually matures and water evaporation exacerbates with temperature, causing RER rarefaction and porosity to increase. Additionally, the dielectric loss factor of air is obviously lower than that of RER, causing an increase in penetration depth.

### 3.4. Microwave Reheating Parameters Determination

#### 3.4.1. Microwave Power Selection

Figure 6 depicts the temperature distribution at the upper (A_1_) and lower (C_1_) layers (Figure 1) of PP rectangular boxes under microwave reheating. With microwave power increasing, the two layers’ average temperatures rise, and layer C_1_ was higher than that of layer A_1_ at the same power and reheating time (Figure 7a). During microwave reheating, the bottom of the microwave oven reflects the microwaves, and the tube wall generates a current through interactions with the electric-magnetic field. This enhances the generation of microwave volumetric heat at the bottom of the RER, resulting in a higher temperature at layer C_1_ than that at layer A_1_.

At a microwave power of 1000 W, the temperature distribution uniformity of layer A_l_ was mediocre due to the obvious difference between the maximum and minimum temperature. Layer C_1_ showed the poorest temperature uniformity at 900 W, and the highest at 800 W (Figure 7b–d). In microwave reheating, the TE wave parallels layers A_1_ and C_1_, had an incidence to the interior of the dielectric material (RER), and dissipated the microwave into volumetric heat. Due to the attenuation of TE wavelength in RER, the electric field distribution in the A_1_ and C_1_ layers form the alternating strength change, with the interval at half wavelength. As a result, the temperatures of A_l_ and C_1_ layers exhibit ‘cold spot’ and ‘hot spot’ phenomena [34]. The higher microwave power caused greater absorption of microwave energy at the RER surface to form the obvious surface heating phenomenon. Electric and magnetic fields produced overheating phenomenon due to the non-resonant scattering effect, which resulted in obvious wall and corner effects with higher temperatures. The greater the power, the lower the uniformity of temperature distribution.

According to Equations (7) and (8), a greater microwave input power ($P_{in}$) causes a larger volume of heat in local areas, increasing temperature difference between high- and low-temperature regions and a decrease in temperature distribution uniformity [35]. With the comparison of temperature distribution under different microwave power, 800 W power was selected as the suitable power for microwave reheating.
(7)$$\Delta Q=\frac{2\pi f\varepsilon^{\prime \prime }P_{in}}{A_{w}c}e^{-2\alpha r}$$
(8)$$\alpha =\frac{2\pi }{\lambda_{0}}{\left[\frac{\varepsilon^{\prime }}{2}\left(\sqrt{1+\left(\frac{\varepsilon^{\prime \prime }}{\varepsilon^{\prime }}\right)-1}\right)\right]}^{\frac{1}{2}}$$
where *f* is microwave frequency (2450 MHz), $P_{in}$ is microwave input power (kW), $A_{w}$ is the cross-sectional area of the waveguide (0.6235 m^2^), *c* represents light velocity (3 × 10^8^ m/s), α is attenuation coefficient, and *r* is material radius (m).

#### 3.4.2. Volume of Lunch Box Selection

The temperature distributions of the A_1_ and C_1_ layers after reheating are shown in Figure 8. As the box volume increases, the average temperature on both the A_1_ and C_1_ layers decreases. Figure 9a illustrates that layer A_1_ has a lower average temperature than layer C_1_. The temperature distribution uniformity of the A_1_ and C_1_ layers are the lowest at 1000 mL and the highest at 650 mL (Figure 9b–d). With an increase in the box volume, the microwave energy dissipated and attenuated in RER and leading to lower penetration depth, where microwave energy dampened before reaching the central area. Due to inadequate microwave energy absorption, the innermost area was heated primarily by heat conduction. Heat transfer and efficiency in RER were lower than that in microwave bulk heat produced, resulting in an increase in material volume and a decrease in temperature distribution uniformity. A volume of 650 mL was selected as the suitable box volume based on a comprehensive comparison.

### 3.5. Temperature Distribution Analysis

#### 3.5.1. Temperature Distribution of Upper and Lower Surfaces

During microwave reheating, there are ‘hot spots’ at the corners and ‘cold spots’ in the center of both the A_1_ and C_1_ layers, indicating an obviously uneven temperature distribution, as shown in Figure 10. As the incident TE plane wave was parallel to the RER boundary layer (along the *x*- and *y*-directions), non-resonant scattering caused electric field concentration in the boundary layer, resulting in the corner concentration [34]. As a result of the microwave energy, energy accumulation at the corner came from the sidewall, upper, and lower layers of RER. Furthermore, the microwave transmitted to the center had been attenuated because the depth of microwave penetration was shorter than the distance from the center to the edge (Figure 5c). In addition, the rising center temperature depended on conduction heating, forming a ‘cold spot’ due to less heat conduction and smaller temperature rise range.

#### 3.5.2. Temperature Distribution of Different Layers

The infrared thermal imager only measured the plane temperature distribution in the *z*-direction of the upper (A_1_) and lower (C_1_) layers, and the measurement had a time delay. The optical fiber temperature sensors were used to measure the real-time temperature data of 13 key points in the A_1_, B_1_, and C_1_ layers, and obtain contour maps of temperature distribution in the *x*-, *y*-, and *z*-directions, in order to observe the temperature changes in RER during microwave reheating.

The *z*-direction (*x*–*y* plane) of the rectangular lunch box was used to generate the temperature distribution contour map of the A_1_, B_1_, and C_1_ layers during microwave reheating of RER, as illustrated in Figure 11. Three layers showed obvious wall and corner effects, and layer C_1_ had a larger ‘hot spot’ area than that in the other two layers. With increases in the reheating time, the ‘hot spot’ area increased gradually, due to microwave reflection on the inner wall of the microwave oven chamber. The energy accumulation at the corner caused the rising rate of temperature. Due to the sample being placed at the center of the glass turntable in the microwave oven, the temperature of layer C_1_ showed an obvious overall heating phenomenon. The microwave reflected on the inner wall of the metal, and partial reflection occurred from the metal at the bottom of the microwave oven to the bottom of the lunch box, resulting in more absorption of microwave energy in the bottom of the sample [34]. Furthermore, the reflected electromagnetic waves lacked a parallel electric field to the lunch box’s boundary and based on the longitudinal magnetic field effect, the magnetic field produced current heating, strengthening the overall heating effect of the lower surface of RER [34].

The contour map of temperature distribution of A_2_, B_2_, C_2_, D_2_, and E_2_ layers under microwave reheating of RER was drawn along the *y*-direction (*x*–*z* plane) of the rectangular lunch box, as shown in Figure 12. There were noticeable thermal aggregation phenomena at the wall and corner and a ‘cold spot’ at the center. The area of the ‘hot spot’ gradually increases as the reheating time increases, while the area of the ‘cold spot’ at the center decreases. In terms of the layers, the largest area of the ‘hot spot’ was in the A_2_ and E_2_ layers, and the largest area of the ‘cold spot’ was in the C_2_ layer.

As illustrated in Figure 13, the *x*-direction (*y*–*z* plane) of the rectangular lunch box was used to generate the temperature distribution contour map of the A_3_, B_3_, C_3_, D_3_, and E_3_ layers in the RER under microwave reheating. The *y*–*z* temperature distribution was similar to the *x*–*z* temperature distribution. ‘Hot spot’ appears at the corner and ‘cold spot’ appears in the center. With increases in the reheating time, RER gradually heated to form the heat transfer from the edge to the interior, resulting in the ‘hot spot’ area increase and the ‘cold spot’ area decrease.

According to the energy conservation law, the microwave volumetric heat generated in RER was consumed via heat accumulation (elevation of temperature), heat conduction (caused by temperature difference), and external heat loss (surface thermal convection and bottom heat conduction of boxes) [34]. The temperature and distribution uniformity of RER gradually increased with the reheating time increasing. The temperature difference caused by the change of position points along *x*- or *y*-coordinates was greater than that along *z*-coordinates. These results were attributed to the internal electric field distribution on the relative position of the RER and the microwave oven waveguide at each discrete position of the RER rotation process [34]. The metal wall of the microwave oven caused the microwave reflection to generate a standing wave, resulting in a multi-mode electric field distribution. According to Fresnel’s law, the multi-mode electric field consisted of TE waves parallel to the surface of the lunch box and TM waves perpendicular to that. Contact between the TE wave and the lunch box’s edge induced microwave scattering and non-resonant reflection, resulting in a strong electric field intensity on the RER at the lunch box’s edge. The TM wave entered into the microwave cavity in a vertical direction from the RER surface (*x*–*y* plane) to create an electric current according to Faraday’s law, generating heat near the food box’s edge. The TE wave enters the RER surface (*x*–*y* plane) and is distributed differentially over the RER’s surface in the form of sine or cosine (depending on the phase of the wave). Various modes of electromagnetic waves were superimposed on the surface of the RER to form an uneven distribution of electric field. The microwave can cause noticeable attenuation as penetration inside the RER. The metal wall at the bottom of the lunch box reflected the electromagnetic wave to generate a longitudinal magnetic field (LSM) since the vertical distance between the bottom of the lunch box and the bottom of the microwave cavity was less than half a wavelength [36]. It was found that the height size of RER impacted temperature dispersion in the vertical direction, but the effect was much lower than that of length and width.

#### 3.5.3. Improvement of Temperature Distribution Uniformity

During microwave reheating, the temperature of RER gradually decreases from the surface to the inside. In a microwave oven, the incidence wave of the TM plane is parallel to the interior wall of the lunch box, creating a non-resonant scattering action that leads to corner overheating. The heat distribution uniformity of temperature caused by electromagnetic waves reflected from the metal wall at the bottom of the microwave oven to the bottom of the lunch box was greater than that produced by plane waves incident on the surface. With the reheating time increasing, the temperature of RER gradually increases, which results in the RER thermal expansion and increase porosity. The dielectric loss of air was significantly lower than that of RER, and the microwave penetration depth gradually increased. The heat conduction effect was caused by the temperature difference in RER to form the heat in the preheated area (‘hot spot’) at the corner transferred to the central area (‘cold spot’). Therefore, reasonably reducing the ‘hot spot’ temperature while increasing the ‘cold spot’ temperature can effectively improve the uniformity of temperature distribution in the microwave reheating process.

Microwave energy is converted into heat energy during propagation in lossy materials (RER), and the conversion capacity is proportional to the loss factor of the materials and the square of the electric field as Equation (9) [35]. When the dielectric loss ($\varepsilon^{\prime \prime }$) of RER in the microwave oven was constant, the higher electric field intensity (*E*) resulted in the faster the temperature elevation.
(9)$$Q=2\pi f\varepsilon_{0}\varepsilon^{\prime \prime }{\left|E\right|}^{2}$$
where *f* is microwave frequency (2450 MHz), $\varepsilon_{0}$ is vacuum dielectric constant (F/m), $\varepsilon^{\prime \prime }$ is relative dielectric loss of RER, and E is electric field strength (V/m).

The non-uniform distribution phenomenon of temperature was related to the distributed uneven electric field inside the RER. To improve the electric field distribution uniformity, the metalized packaging was designed with microwave-active materials. Altering the microwave transmission mode and energy absorption method can improve the uniformity of the electric field distribution. The inductive current was generated when a microwave passed through metal packaging materials with high conductivity to generate heat energy. When electromagnetic waves passed through the high dielectric loss materials, the materials absorbed microwave energy.

### 3.6. Metalized Packaging Structure Design

To improve the energy distribution of the electromagnetic field in the packaging, a metalized packaging structure was designed. It makes the electromagnetic field distribution reasonable according to the expected position, realizing different parts for differential heating [34]. The metalized packaging structure was developed based on the temperature distribution of RER during microwave reheating in the *x*-, *y*-, and *z*-directions, where *x* and *y* were the long and short sides of the lunch box, and *z* was the direction of TE transverse wave propagation.

The metalized packaging structure was developed according to the temperature distribution contour maps (in *x*-, *y*-, and *z*-directions) at the 180-s reheating time. Given the temperature distribution of RER undergoing reheating showed obvious wall and corner effects, the metalized packaging structure reflects microwaves at the ‘hot spot’ and absorbs them at the secondary ‘hot spot’, which decreases the ‘hot spot’ and increases the ‘cold spot’ temperature, thus improving the uniformity of temperature distribution [34]. Uneven and discontinuous surfaces, as well as rectangular containers with a radius greater than 20 mm, can reduce the wall and corner effect [34]. Metalized packaging film creates an uneven plane, and metalized packaging structure design can create a rounded rectangle [34]. The temperature distribution contour maps for metalized packaging structure reference are shown in Figure 14. The box cover structure refers to the upper layers (A_1_) of the *x*–*y* plane, the box bottom structure refers to the lower layers (C_1_) of the *x–y* plane, the box length–height side face structure refers to the *x*–*z* plane’s outer layer (A_2_), and the box width–height side face structure refers to the *y*–*z* plane’s outer layer (A_3_).

In the contour map of temperature distribution, the ‘hot spots’ were marked with ① symbol, and the secondary ‘hot spots’ were marked with ② symbols. Microwave active materials were used at symbols ① and ②, and the effect on the microwave at symbol ① was greater than that at symbol ②. Microwave active packaging was constructed from microwave active materials and could change the electric field distribution in RER under microwave heating. Designing proper active packaging can improve the microwave heating uniformity of RER. Microwave active packaging’s impact on microwaves included shielding, field adjustment, and sensory absorption. Metalized packaging materials with high conductivity generate current and transform heat energy in the magnetic field (field adjustment), while high dielectric loss materials absorb microwave energy and convert it to heat loss (sensory absorption), allowing the two characteristics to weaken (‘hot spot’ location) or enhance (‘cold spot’ location) the intensity of microwave energy incident to RER. The heat was transferred into the adjacent samples via heat conduction for packaging active material heated, and the sample at the ‘hot spot’ location was heated slowly due to less heat conduction. The ‘cold spot’ location of packaging was devoid of microwave active material, the microwave was absorbed and heated in a microwave cavity via multiple reflections incident at the ‘cold spot’ location. The wall and corner effect can be obviously reduced, and the temperature distribution uniformity of RER improved by lessening the temperature rise of the ‘hot spot’ location and accelerating the temperature rise of the ‘cold spot’ location.

The volume and thickness of the spray film are indicated in Table 2 while applying the film solution at symbols ① and ② of the metallization packaging for RER. Figure 15 indicates the A1 and C1 layers’ temperature distribution of RER packed in metallization packaging under microwave reheating.

Compared with un-metalized packaging I, in metalized packaging II and III, the ‘hot spot’ in the corner of the A_1_ layer vanished while the ‘cold spot’ in the C_1_ layer remained but at a different location. In metalized packing IV, the ‘hot and cold spot’ in the A_1_ layer remained present but vanished in the C_1_ layer. The area and position of the ‘hot spot’ and ‘cold spot’ formed in the microwave-reheating process of RER with the same structure and different metalized packaging thickness were different. This was related to the microwave reflection and scattering process by metalized packaging.

The average temperature, temperature uniformity coefficient, maximum temperature, and minimum temperature of the A_1_ and C_1_ layers were compared to evaluate the microwave reheating effect of the four metalized packaging methods (Figure 16). Compared with un-metalized packaging I, the A_1_ layer’s average temperature in metalized packaging II and IV decreased, while metalized packaging III increased. The average temperature of the C_1_ layer decreased in metalized packaging II and III; however, it increased in metalized packaging IV. Compared with un-metalized packaging I, metalized packaging II failed to enhance the temperature distribution uniformity of RER undergoing reheating, whereas metalized packaging III improved uniformity to a certain extent, and metalized packaging IV obviously improved uniformity to some extent. The maximum and minimum temperature differences between the A_1_ and C_1_ layers are listed in the following order: metalized packaging II, metalized packaging III, and metalized packaging IV.

After comprehensively comparing the microwave reheating effects of four metalized packaging forms, the highest average temperature of RER was found in metalized packaging III, and metalized packaging IV showed the highest temperature distribution uniformity. As a result, metalized packaging IV is chosen as the RER’s metalized packaging.

As the models and experiments revealed, the rectangular mashed potato also showed temperature non-uniformity during microwave reheating, with ‘hot spots’ at the edge and ‘cold spots’ at the center [37,38]. The microwave active packaging can increase the distribution uniformity of the electric field in the frozen pie, which can not only heat faster but also noticeably improves temperature uniformity [39]. The design of the metalized packaging structure in this paper can regulate the distribution of field intensity inside RER, improve the distribution uniformity of the electric field, and improve the effect of microwave reheating. The ability to enhance temperature distribution uniformity offers the opportunity to dramatically reduce cooking time compared with conventional oven cooking while achieving similar or even improved cooking performance.

## 4. Conclusions

Metalized RER packaging was constructed using an aluminum film solution, and the aluminum film pattern was sprayed at 3.5 × 10^−4^ mL/mm^2^, with an aluminum film thickness of 0.03 mm. When a microwave incident on metalized packaging, the aluminum film alters the way the electromagnetic wave incident into the lunch box as well as the distribution of the electromagnetic field inside the microwave cavity, thus effectively avoiding the appearance of electromagnetic field intensity in the lunch box. As a result, this improves the uniformity of the electromagnetic field and temperature distribution, and decreases the reheating time. The influence on taste and texture is minimal due to the high heating speed and low water loss. Microwave heating is a common heating method that requires no new equipment and has no potential cost. The approach is viable as microwave heating metalized packaged food offers the features of convenience, speed, high heating efficiency, and good heating effect.

Shielding and field modification will continue to be valuable tools for microwave package designers, and this interest will drive the further evolution of structures and manufacturing methods. The development of packages with more complex shapes to accommodate multi-component instant meals represents a potential future research path. Subsequent experiments may optimize the metalized packaging structure to achieve the desired microwave reheating effect. The results of this study provide a theoretical foundation and technological support for the industrialization of scale production and packaging of prepared food in the ‘Central Kitchen’.

## Figures and Tables

**Figure 1 foods-12-02938-f001:**
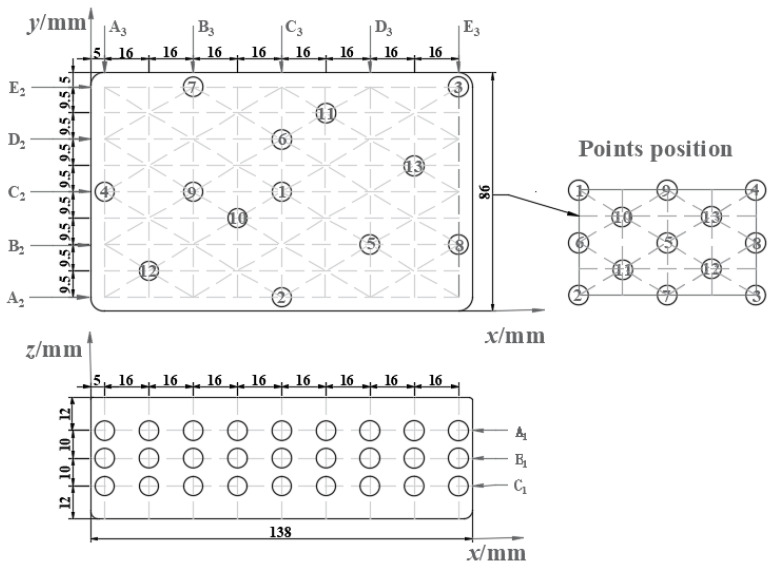
Arrangement scheme of measurement points.

**Figure 2 foods-12-02938-f002:**
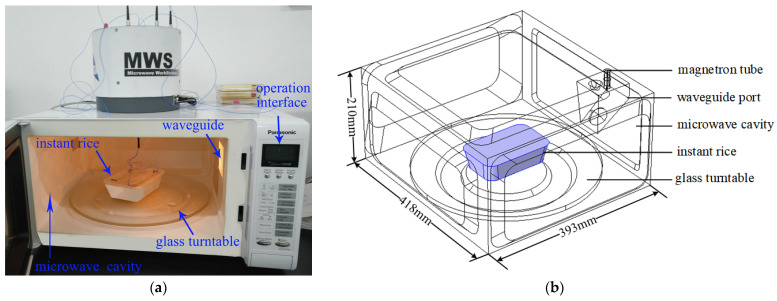
Schematic diagram of a geometric model of microwave reheating of refrigerated rectangular RER. (**a**) Photograph of the microwave reheating RER setup; (**b**) the simplified geometric model of microwave reheating RER. The microwave workstation’s frequency is 2.45 GHz, and the temperature measurement ranged from −40 °C to 250 °C.

**Figure 3 foods-12-02938-f003:**
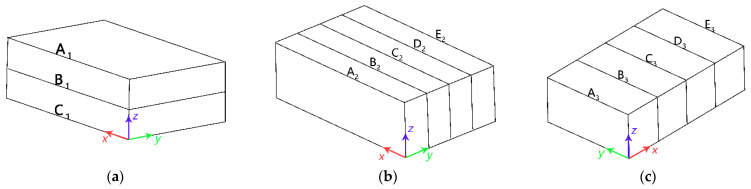
Sketch map of drawing contour map for temperature distribution. (**a**) The sketch map of the drawing contour map for the temperature distribution of *x*–*y* plane in *z* direction; (**b**) sketch map of the drawing contour map for the temperature distribution of *x*–*z* plane in *y* direction; (**c**) sketch map of drawing contour map for temperature distribution of *y*–*z* plane in *x* direction.

**Figure 4 foods-12-02938-f004:**
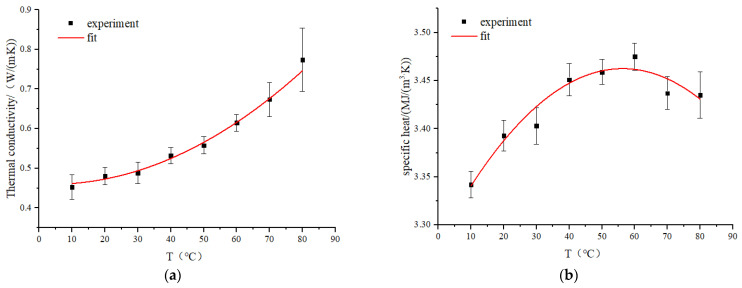
The thermal characteristics of RER under different temperatures, and the RER’s moisture content is 63.23 ± 0.87%. (**a**) The thermal conductivity of RER; (**b**) the specific heat of RER.

**Figure 5 foods-12-02938-f005:**
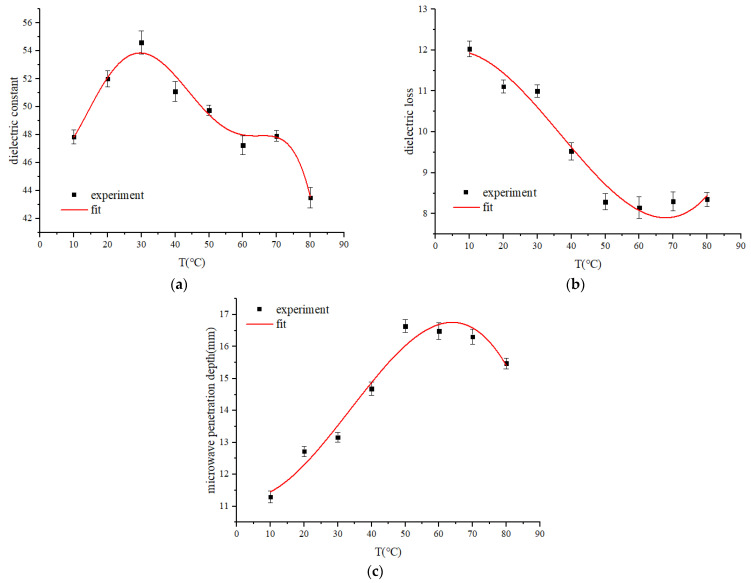
The dielectric properties of RER under different temperatures, the RER’s moisture content is 63.23 ± 0.87%, and the measurement frequency is 2.45 GHz. (**a**) The dielectric constant of RER; (**b**) the dielectric loss factor of RER; (**c**) microwave penetration depth inside RER.

**Figure 6 foods-12-02938-f006:**
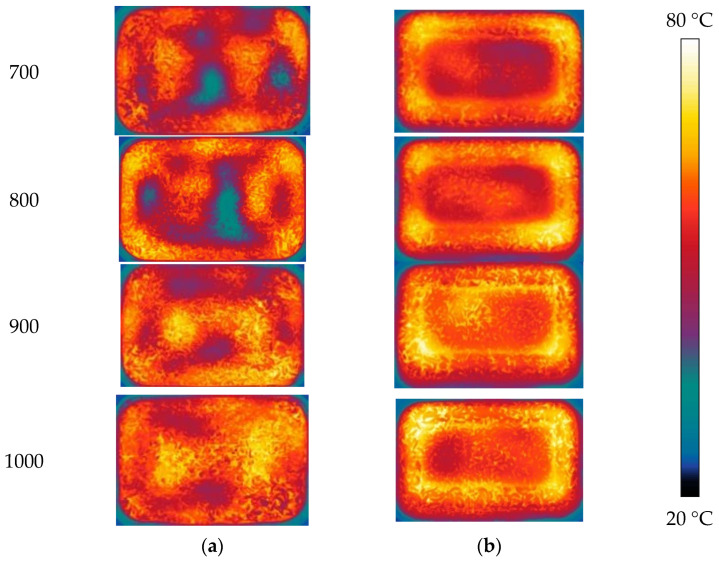
Temperature distribution after microwave reheating with different microwave powers. The PP rectangular boxes’ bottom sizes are 138 × 86 mm^2^, and heights are 44 mm. The microwave powers are 700, 800, 900, and 1000 W, respectively; reheating time is 180 s, and the measurement frequency is 2.45 GHz. The RER’s moisture content is 63.23 ± 0.87%. (**a**) The temperature distribution in layer A_1_; (**b**) the temperature distribution in layer C_1_.

**Figure 7 foods-12-02938-f007:**
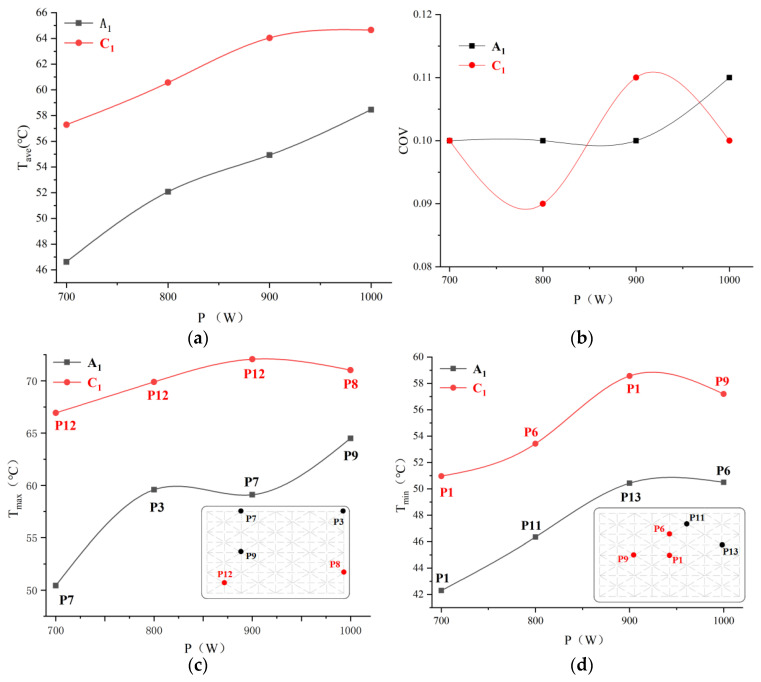
Temperature values after reheating with different microwave powers. (**a**) Average temperature of RER under different power; (**b**) temperature distribution uniformity of RER under different power; (**c**) maximum temperature of RER under different power; (**d**) minimum temperature of RER under different power.

**Figure 8 foods-12-02938-f008:**
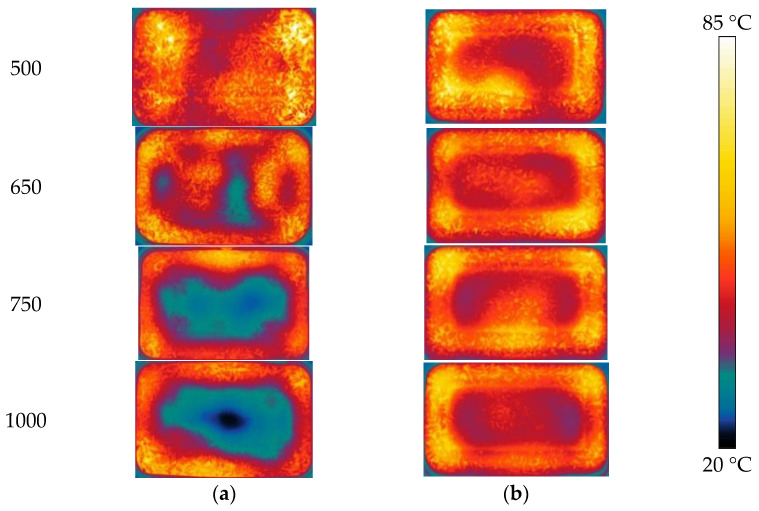
Temperature distribution of RER with different volumes under microwave reheating. The PP rectangular boxes’ bottom sizes are 138 × 86 mm^2^, and heights are 40, 44, 50, and 65 mm, respectively. We used a microwave power of 800 W, reheating time of 180 s, and measurement frequency of 2.45 GHz. The RER’s moisture content is 63.23 ± 0.87%. (**a**) The temperature distribution in layer A_1_ of RER; (**b**) the temperature distribution in layer C_1_ of RER.

**Figure 9 foods-12-02938-f009:**
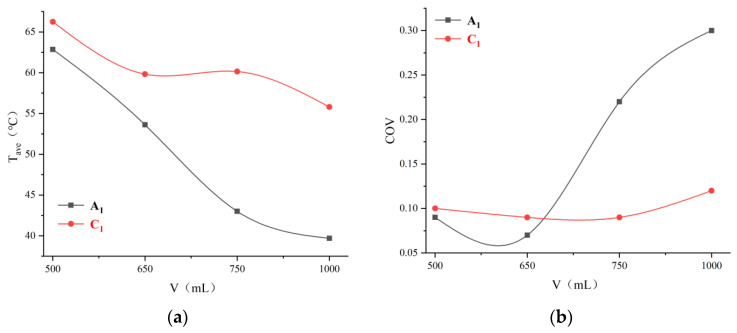
Temperature values after reheating with different volumes. (**a**) Average temperature of RER under different volumes; (**b**) temperature distribution uniformity of RER under different volumes; (**c**) maximum temperature of RER under different volume volumes; (**d**) minimum temperature of RER under different volumes.

**Figure 10 foods-12-02938-f010:**
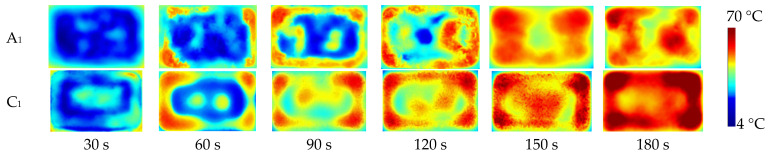
Temperature distribution of RER during reheating. The RER’s moisture content is 63.23 ± 0.87%. The PP rectangular boxes’ bottom sizes are 138 × 86 mm^2^, and heights are 44 mm. We used a microwave power of 800 W, reheating time of 180 s, and measurement frequency of 2.45 GHz.

**Figure 11 foods-12-02938-f011:**
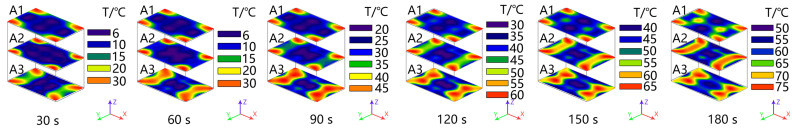
Contour map of temperature distribution on the *x*–*y* plane of RER during reheating. The RER’s moisture content is 63.23 ± 0.87%. The PP rectangular boxes’ bottom sizes are 138 × 86 mm^2^, and heights are 44 mm. We used a microwave power of 800 W, reheating time of 180 s, and measurement frequency of 2.45 GHz.

**Figure 12 foods-12-02938-f012:**
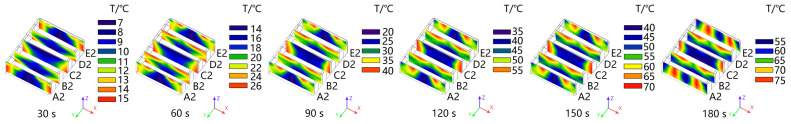
Contour map of temperature distribution on the *x*–*z* plane of RER during reheating. The RER’s moisture content is 63.23 ± 0.87%. The PP rectangular boxes’ bottom sizes are 138 × 86 mm^2^, and heights are 44 mm. We used a microwave power of 800 W, reheating time of 180 s, and measurement frequency of 2.45 GHz.

**Figure 13 foods-12-02938-f013:**
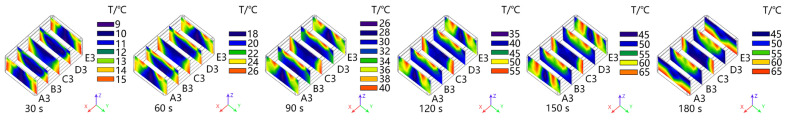
Contour map of temperature distribution on the *y*–*z* plane of RER during reheating. The RER’s moisture content is 63.23 ± 0.87%. The PP rectangular boxes’ bottom sizes are 138 × 86 mm^2^, and heights are 44 mm. We used a microwave power of 800 W, reheating time of 180 s, and measurement frequency of 2.45 GHz.

**Figure 14 foods-12-02938-f014:**
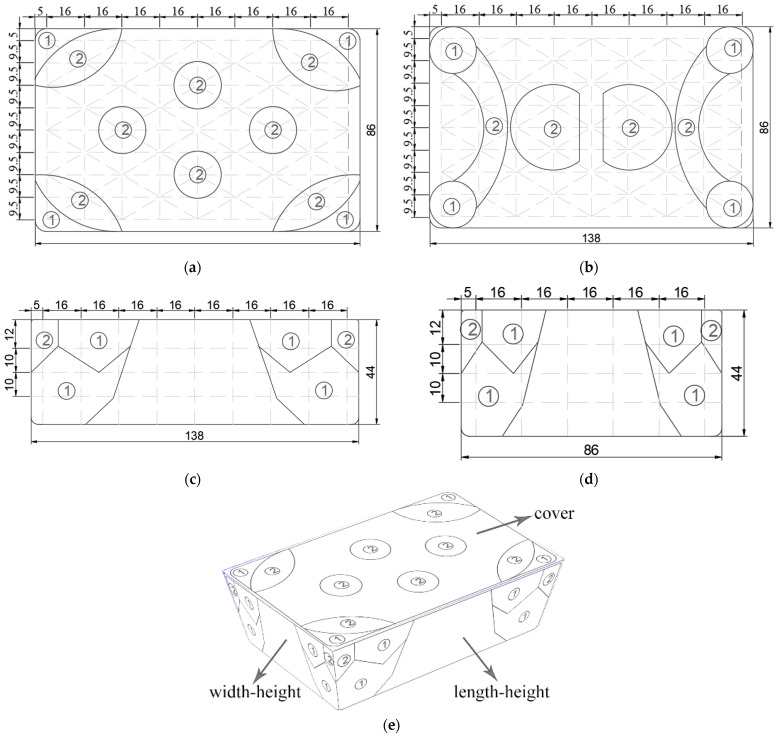
Structure diagram of metalized packaging for RER. (**a**) Structure diagram in the cover face of metallization packaging; (**b**) structure diagram in the bottom face of metalized packaging; (**c**) structure diagram in the length–height side face of metalized packaging; (**d**) structure diagram in the width–height side face of metalized packaging; (**e**) structure diagram on the cubic effect of metalized packaging. unit: mm.

**Figure 15 foods-12-02938-f015:**
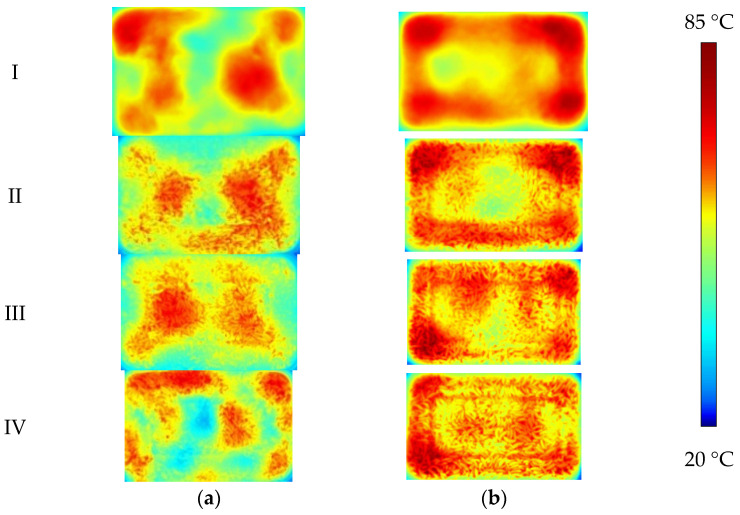
Temperature distribution of RER with metalized packaging after microwave heating. The RER’s moisture content is 63.23 ± 0.87%. The PP rectangular boxes’ bottom sizes are 138 × 86 mm^2^, and heights are 44 mm. We used a microwave power of 800 W, reheating time of 180 s, and measurement frequency of 2.45 GHz. (**a**) The temperature distribution in layer A_1_ of RER; (**b**) the temperature distribution in layer C_1_ of RER.

**Figure 16 foods-12-02938-f016:**
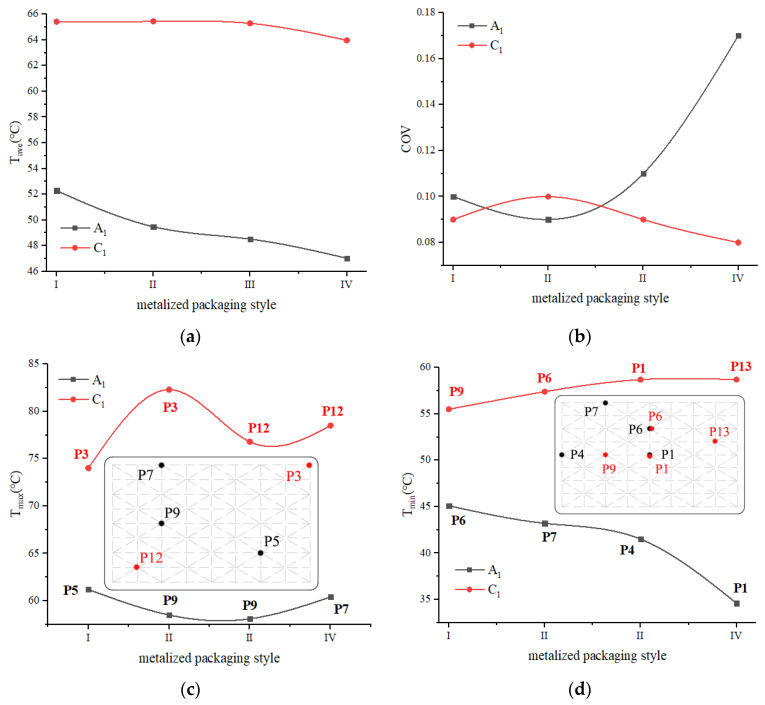
Temperature distribution uniformity of RER with metalized packaging. (**a**) Average temperature of RER with metalized packaging after heating; (**b**) temperature distribution uniformity of RER with metalized packaging after heating; (**c**) maximum temperature of RER with metalized packaging after heating; (**d**) minimum temperature of RER with metalized packaging after heating.

**Table 1 foods-12-02938-t001:** Moisture content (%) of RER under different positions, reheating times, and 800 W microwave power.

Locations	Reheating Times (s)					
	30	60	90	120	150	180
A_1_	61.52 ± 1.46	64.76 ± 1.72	63.17 ± 2.06	62.68 ± 2.60	64.98 ± 0.40	62.71 ± 1.01
B_1_	64.10 ± 2.30	65.83 ± 3.11	63.26 ± 1.38	61.89 ± 2.70	63.49 ± 2.46	63.50 ± 1.23
C_1_	66.86 ± 1.76	68.05 ± 1.33	65.16 ± 0.47	67.26 ± 2.43	64.66 ± 0.78	66.84 ± 2.07

**Table 2 foods-12-02938-t002:** Preparation of metalized packaging film for RER.

Metalized Packaging	$\mathrm{Volume}\;(\mathrm{mL}/{mm}^{2})$		Thickness (mm)	
	Symbol ①	Symbol ②	Symbol ①	Symbol ②
I	0	0	0	0
II	$7.0\times 10^{-4}$	$3.5\times 10^{-4}$	0.60	0.30
III	$7.0\times 10^{-4}$	$7.0\times 10^{-4}$	0.60	0.60
IV	$3.5\times 10^{-4}$	$3.5\times 10^{-4}$	0.30	0.30

The symbols ① and ② in the table are shown in Figure 14.

## Data Availability

The datasets generated for this study are available on request to the corresponding author.

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
