# Peer review of "Improvement of Temperature Distribution Uniformity of Ready-to-Eat Rice during Microwave Reheating via Optimizing Packaging Structure"

_foods, 2023, doi:10.3390/foods12152938_

Round 1
Author Response
Responses Letter
Dear Ms. Knezevic,
Thank you for your letter and give us an opportunity to revise our manuscript entitled “Improvement of Temperature Distribution Uniformity of Ready-to-eat Rice during Microwave Reheating via Optimizing Packaging Structure”(Manuscript Number: foods-2517487). Those comments from reviewers are all valuable and very helpful for revising and improving our paper, as well as the important guiding significance to our researches. We have studied comments carefully and have made corrections which we hope to meet with approval.
We have outlined every change made in response to reviewers’ comments and provide suitable revisions in the revised manuscript and not only in the response letter. The responds to the reviewer’s comments are marked in red, and other changes are marked in blue in the revised manuscript.
Kind regards,
Xianzhe Zheng, PhD
28-july-2023
Corresponding author: Xianzhe Zheng
E-mail: zhengxz@neau.edu.cn
The specific revisions and list of responses of manuscript are as follows:
Responds to the Reviewer #1:
Comments:
The manuscript entitled "Improvement of temperature distribution uniformity of instant rice during microwave reheating via optimizing packaging structure" is well organized and written in a comprehensive way and contains good information. Therefore, this can be accepted for publication in Foods. However, some modifications with regard to the following questions and comments should be made in order to improve its quality.
Response: Thanks for your positive comments and constructive suggestions on manuscript. According to your comments and suggestions, we have carefully and thoroughly revised the manuscript. The specific responses to the comments by point to point are as follows.
Introduction:
(1) The section provides a clear and concise overview of the importance of microwave heating technology in food processing and the potential issues associated with microwave reheating. However, the highlighted food sample used in this work, instant rice, was not mentioned in this section. The authors should briefly address some information of the sample and some problems related to the sample.
Response: We are very sorry for “the highlighted food sample used in this work, instant rice, was not mentioned in this section”. we add the “information of the sample” and marked it in red font in the revised manuscript as shown in Line 60-64.
(2) Generally, instant rice is a white rice that is partly precooked and then is dehydrated and packed in a dried form similar in appearance to that of regular white rice. Please use appropriate word for the tested sample throughout the manuscript. ‘Convenient rice’ or ‘Ready-to-eat rice’ could be possible.
Response: Thanks for your comment and suggestion. We have corrected the “instant rice” to “Ready-to-eat rice” and marked it in red font in the revised manuscript.
Materials and Methods:
(1) Provide a more detailed explanation of the experimental setup, including the type of microwave oven used, the power level and duration of the microwave heating, and the temperature measurement method. This will help readers better understand the experimental process and the validity of the results.
Response: Thanks for your valuable comment. We add the “detailed explanation of the experimental setup” and marked in red font in the revised manuscript as shown in Line 162-175.
The revised portion:
“2.2.9. Measurement Method of Temperature Data
As illustrated in Figure 2, microwave reheating began after placing the RER (with the volume of 500, 650, 750, and 1000 mL, respectively) into the microwave workstation (the frequency is 2.45 GHz) and setting the microwave power (of 700, 800, 900, and 1000 W) and heating time (for 30, 60, 90, 120, 150, and 180 s). The real-time temperature of key points is acquired at 0.6 s intervals using an optical fiber temperature sensor (FOT-L-SD-C1-F1-M2-R1-ST, FLIR Inc., USA) and FISO software (MWS, FLIR Inc., USA). The temperature distribution of layers A1, B1, and C1 were measured using an infrared thermal imager (FLIR E95, FLIR Inc., USA) at 30, 60, 90, 120, 150, and 180 s of reheating time, respectively.
|
|
(a) |
(b) |
Figure 2. Schematic diagram of a geometric model of microwave reheating of refrigerated rectangular RER. (a) The object picture of microwave reheating RER; (b) the simplified geometric model of microwave reheating RER. The microwave workstation’s frequency is 2.45 GHz, and the temperature measurement range is -40 oC to 250 oC.”.
(2) Provide a more detailed explanation of the sample preparation process, including the selection of instant rice including its variety. This will help readers understand the sample characteristics and the potential impact of sample preparation on the experimental results.
Response: Thanks for your comment and suggestion. We add the “detailed explanation of the sample preparation process” and marked in red font in the revised manuscript as shown in Line 98-105.
(3) Provide a more detailed explanation of all parameter measurements, not just referring the references. This will help readers follow and understand clearly.
Response: Thanks for your valuable suggestion. We add the “detailed explanation of all parameter measurements” and marked in red font in the revised manuscript as shown in Line 108-114, Line 117-118, Line 130-135, and Line 140-148.
(4) Provide a brief explanation of importance of all parameters with relation to the significance of the problem.
Response: Thank you for your valuable comments. We add the “brief explanation of importance of all parameters with relation to the significance of the problem” and marked in red font in the revised manuscript as shown in Line 65-80.
The revised portion:
“Food absorbs microwave energy and then transforms into heat energy during the microwave reheating process. Heat energy raises the temperature of food and facilitates water diffusion and evaporation. Food properties change affect the temperature distribution during the reheating process, and dielectric properties determine food's ability to absorb and transform microwave energy [22]. Studying the change of food dielectric properties is helpful to analyze the influence of microwave energy absorption on temperature distribution. In the process of microwave reheating, changes in thermal properties can indicate the law of heat consumption and transport. Food characteristic indexes are useful for clarifying the heat and mass transfer mechanism of food during microwave reheating, analyzing the causes of uneven temperature distribution, and providing a reasonable basis for improving uneven temperature distribution. Furthermore, the active packaging has the ability to change the features of the electric field distribution in the RER under microwave heating, which is possible to build a realistic active packaging to improve the uniformity of microwave heating of RER [5]. Microwave food packaging, as an efficient and low-cost means of improving microwave heating uniformity, offers vast application potential in the microwave food business.”.
(5) Provide a more detailed explanation of the data analysis methods used to evaluate the temperature distribution uniformity of instant rice, including the statistical tests used to compare the results of different packaging structures. This will help readers understand the validity of the results and the significance of the proposed solution.
Response: Thanks for your valuable suggestion. We add the “more detailed explanation of the data analysis methods used to evaluate the temperature distribution uniformity of instant rice” and marked in red font in the revised manuscript as shown in Line 186-187.
Results an Analysis:
(1) Provide a more detailed explanation of the factors that influence the temperature distribution uniformity of instant rice during microwave reheating, including the impact of the shape and size of the lunch box, the moisture content of the rice, and the power level and duration of the microwave heating. This will help readers understand the complexity of the problem and the potential limitations of the proposed solution.
Response: Thank you for your valuable comments. We add the “more detailed explanation of the factors that influence the temperature distribution uniformity of instant rice during microwave reheating” and marked in red font in the revised manuscript as shown in Line 304-308 (Figure 6.), 337-341 (Figure 8.), 366-369 (Figure 10.), 386-389 (Figure 11.), 399-402 (Figure 12.), 411-414 (Figure 13.), 528-532 (Figure 15.).
(2) Consider including a discussion of the potential practical applications of the proposed solution, including the feasibility of implementing the metalized packaging structure in commercial instant rice products. This will help readers understand the potential impact of the study on the food industry and consumer products.
Response: Thank you for your valuable comments. We add the “discussion of the potential practical applications of the proposed solution” and marked in red font in the revised manuscript as shown in Line 567-572.
(3) Provide a more detailed explanation of densities and thermal characteristics of instant rice as affected by operating factors used in this work. This will help readers understand benefits of this information.
Response: Thanks for your valuable suggestion. We add the “detailed explanation of densities and thermal characteristics of instant rice as affected by operating factors used in this work” and marked in red font in the revised manuscript as shown in Line 255-256 (Figure 4.), Line 277-279 (Figure 5.).
Conclusions:
(1) Provide a more detailed summary of the key findings of the study, including the impact of the metalized packaging structure on the temperature distribution uniformity of instant rice during microwave reheating.
Response: Thank you for your valuable suggestion. We add the “detailed summary of the key findings of the study” and marked in red font in the revised manuscript as shown in Line 574-576.
(2) Consider including a discussion of the potential limitations of the proposed solution, including the impact of the packaging structure on the taste and texture of the instant rice, as well as the potential cost and feasibility of implementing the solution in commercial products.
Response: Thank you for your valuable comments. We add the “discussion of the potential limitations of the proposed solution” and marked in red font in the revised manuscript as shown in Line 581-585.
(3) Provide a more detailed explanation of the underlying physical principles that govern the temperature distribution uniformity of instant rice during microwave reheating, including the role of dielectric properties and the impact of electromagnetic field distribution.
Response: Thank you for your valuable comments. We add the “detailed explanation of the underlying physical principles that govern the temperature distribution uniformity of instant rice during microwave reheating” and marked in red font in the revised manuscript as shown in Line 576-582.
- Consider including a discussion of the potential future directions for research in this area, including the development of new packaging materials and the optimization of microwave heating parameters to achieve better temperature distribution uniformity.
Response: Thanks for your valuable suggestion. We add the “discussion of the potential future directions for research in this area” and marked in red font in the revised manuscript as shown in Line 586-589.
The other changes are marked in blue in the revised manuscript
We tried our best to improve the manuscript and made some other changes marked in blue in the revised manuscript. These changes will not depart from the main objectives and frameworks of the manuscript. We appreciate for Editors/Reviewers’ warm work earnestly, and hope that the revisions will meet with approval.
Once again, thank you very much for your comments and suggestions.

Reviewer 2 Report
In presented manuscript titled “Improvement of Temperature Distribution Uniformity of Instant Rice during Microwave Reheating via Optimizing Packaging Structure” the authors carried out comprehensive research with usage of different techniques and methods, but missed very important part with statement of purpose of the work, and discussing how their work will be useful for science and industry. Thus, the text should be modified to make it more attractive to readers. At presented state the manuscript requires major revision. The main comments and recommendations are listed below.
What is a purpose of the work? This should be pointed out in the end of Introduction
Equations 1 and 2 are well known and generally used, so they are not necessary to show in the text.
All details for equipment, software and chemicals should be given in the text (Manufacturer, Localization including city and country).
There are many figures in the text - some of them can be combined
Discussion part should be supported by more recent and relevant data
Conclusion should be supported by data obtained.
Future prospects should be discussed in the end of Conclusions. How the results obtained will be useful for the authors and other researchers. Maybe the results will be transferred directly to industry. This part should be clarified. At this moment the significance of the work can not be assessed.
English should be checked for typos and grammatical errors
English should be checked for typos and grammatical errors
Author Response
Responses Letter
Dear Ms. Knezevic,
Thank you for your letter and give us an opportunity to revise our manuscript entitled “Improvement of Temperature Distribution Uniformity of Ready-to-eat Rice during Microwave Reheating via Optimizing Packaging Structure”(Manuscript Number: foods-2517487). Those comments from reviewers are all valuable and very helpful for revising and improving our paper, as well as the important guiding significance to our researches. We have studied comments carefully and have made corrections which we hope to meet with approval.
We have outlined every change made in response to reviewers’ comments and provide suitable revisions in the revised manuscript and not only in the response letter. The responds to the reviewer’s comments are marked in red, and other changes are marked in blue in the revised manuscript.
Kind regards,
Xianzhe Zheng, PhD
28-july-2023
Corresponding author: Xianzhe Zheng
E-mail: zhengxz@neau.edu.cn
The specific revisions and list of responses of manuscript are as follows:
Responds to the Reviewer #2:
Comment:
In presented manuscript titled “Improvement of Temperature Distribution Uniformity of Instant Rice during Microwave Reheating via Optimizing Packaging Structure” the authors carried out comprehensive research with usage of different techniques and methods, but missed very important part with statement of purpose of the work, and discussing how their work will be useful for science and industry. Thus, the text should be modified to make it more attractive to readers. At presented state the manuscript requires major revision. The main comments and recommendations are listed below.
Response: Thanks for your positive comments and constructive suggestions on manuscript. According to your comments and suggestions, we have carefully and thoroughly revised the manuscript. The specific responses to the comments by point to point are as follows.
(1) What is a purpose of the work? This should be pointed out in the end of Introduction
Response: We are sorry that we did not highlight clearly the purpose of the work. We add the “purpose of the work” and marked in red font in the revised manuscript as shown in Line 81-89.
(2) Equations 1 and 2 are well known and generally used, so they are not necessary to show in the text.
Response: Thanks for your comment and suggestion. We have deleted Equations 1 and 2.
(3) All details for equipment, software and chemicals should be given in the text (Manufacturer, Localization including city and country).
Response: Thank you for your comment and suggestion. We add the “details for equipment, software and chemicals” and marked in red font in the revised manuscript as shown in Line 99-103, Line 127, Line 132, Line 134, Line 137, Line 146, Line 163-171, and Line 218-223.
(4) There are many figures in the text - some of them can be combined
Response: Thanks for your valuable comment and suggestion. We combine the figures and marked the figure serial number in red font in the revised manuscript as shown in Line 255-256 (Figure 4), Line 277-279 (Figure 5).
(5) Discussion part should be supported by more recent and relevant data
Response: Thank you for your valuable comment and suggestion. We add the “Discussion part that supported by more recent and relevant data” and marked in red font in the revised manuscript as shown in Line 563-572.
(6) Conclusion should be supported by data obtained.
Response: Thanks for your valuable comment and suggestion. We check and correct the conclusion, and marked in red font.
(7) Future prospects should be discussed in the end of Conclusions. How the results obtained will be useful for the authors and other researchers. Maybe the results will be transferred directly to industry. This part should be clarified. At this moment the significance of the work can not be assessed.
Response: Thank you for your valuable comment and suggestion. We add the discussion “How the results obtained will be useful for the authors and other researchers” and marked in red font in the revised manuscript as shown in Line 591-593.
(8) English should be checked for typos and grammatical errors
Response: Thanks for your suggestions. We have checked for typos and grammatical errors and marked in blue font in the revised manuscript.
The other changes are marked in blue in the revised manuscript
We tried our best to improve the manuscript and made some other changes marked in blue in the revised manuscript. These changes will not depart from the main objectives and frameworks of the manuscript. We appreciate for Editors/Reviewers’ warm work earnestly, and hope that the revisions will meet with approval.
Once again, thank you very much for your comments and suggestions.

Reviewer 3 Report
Microwave is wide scale used for the preparation of instant food. Temperature distribution plays crucial role in the efficiency of heating by microwave irradiation. The temperature (in)homogeneity of irradiated system is depended on many parameters: the dielectric parameters, penetration depth, frequency of irradiation, components and physicochemical state of the processed system and the sample geometry, for examples. In the case of packaged foods the structure and dielectric behaviour of packaging materials have also effect on the heating efficiency and temperature distribution. The manuscript foods-2517487 deals with the temperature distribution uniformity of microwave irradiation using packaged instant rice as model system focusing on the determination of appropriate metalized package structure. Therefore, the topic of the manuscript can be considered as relevant and it can provide useful data and information for the readers. The manuscript has a logic structure, but Introduction section is too superficial. Some parts of methodology section need also revision to make it clear (see my comments). Although manuscript contains novel and interesting results (that have relevance for the practice, as well) it need thoroughly revision to make it more complete and clear.
Comments, suggestions:
[1.] Please highlight clearly the novelties of the study in the Introduction section.
[2.] Please explain why instant rice was used as model system for the research.
[3.] In my opinion, the Introduction section is too superficial. Please discuss the role of dielectric parameters in temperature distribution and heating efficiency in the Introduction section, as well.
[4.] In line 90-91 check the parameters for calculation Eq.(3). (difference between the densities).
[5.] Please give the type of thermal characteristic analyser and condition of measurements (section 2.2.5).
[6.] Please provide the details of dielectric measurements (analyser, conditions etc.).
[7.] In line 105 wavelength is mentioned, but is not used in Eq. (4).
[8.] Please provide the details of microwave unit used for heating (frequency, positioning of packaged system etc.).
[9.] The dielectric constant and dielectric loss factor (and therefore the penetration depth) is also frequency dependent. Please provide the measuring frequency for these data (Figure 5,6,7).
[10.] Establishments in section 3.5.1 need to be strengthened by references, as well.
[11.] Please add reference(s) to line 361-369.
[12.] Please check and correct the use of sub/superscript in the MS (see line 264, for instance).
[13.] The visibility of Figure 13-15 is very poor. Please improve it.
[14.] Results related to metal packaging structure design (section 3.6) need discussion with relevant references, as well.
Author Response
Responses Letter
Dear Ms. Knezevic,
Thank you for your letter and give us an opportunity to revise our manuscript entitled “Improvement of Temperature Distribution Uniformity of Ready-to-eat Rice during Microwave Reheating via Optimizing Packaging Structure”(Manuscript Number: foods-2517487). Those comments from reviewers are all valuable and very helpful for revising and improving our paper, as well as the important guiding significance to our researches. We have studied comments carefully and have made corrections which we hope to meet with approval.
We have outlined every change made in response to reviewers’ comments and provide suitable revisions in the revised manuscript and not only in the response letter. The responds to the reviewer’s comments are marked in red, and other changes are marked in blue in the revised manuscript.
Kind regards,
Xianzhe Zheng, PhD
28-july-2023
Corresponding author: Xianzhe Zheng
E-mail: zhengxz@neau.edu.cn
The specific revisions and list of responses of manuscript are as follows:
Responds to the Reviewer #3:
Comment:
Microwave is wide scale used for the preparation of instant food. Temperature distribution plays crucial role in the efficiency of heating by microwave irradiation. The temperature (in)homogeneity of irradiated system is depended on many parameters: the dielectric parameters, penetration depth, frequency of irradiation, components and physicochemical state of the processed system and the sample geometry, for examples. In the case of packaged foods the structure and dielectric behaviour of packaging materials have also effect on the heating efficiency and temperature distribution. The manuscript foods-2517487 deals with the temperature distribution uniformity of microwave irradiation using packaged instant rice as model system focusing on the determination of appropriate metalized package structure. Therefore, the topic of the manuscript can be considered as relevant and it can provide useful data and information for the readers. The manuscript has a logic structure, but Introduction section is too superficial. Some parts of methodology section need also revision to make it clear (see my comments). Although manuscript contains novel and interesting results (that have relevance for the practice, as well) it need thoroughly revision to make it more complete and clear.
Response: Thanks for your positive comments and constructive suggestions on manuscript. According to your comments and suggestions, we have carefully and thoroughly revised the manuscript. The specific responses to the comments by point to point are as follows.
- Please highlight clearly the novelties of the study in the Introduction section.
Response: We are sorry that we did not highlight clearly the novelties of the study. The revised portion “the active packaging has the ability to change the features of the electric field distribution in the RER under microwave heating, which is possible to build a realistic active packaging to improve the uniformity of microwave heating of RER [5]. Microwave food packaging, as an efficient and low-cost means of improving microwave heating uniformity, offers vast application potential in the microwave food business” has been marked in red font in the revised manuscript as shown in Line 76-80.
(2) Please explain why instant rice was used as model system for the research.
Response: We are very sorry for our unclear expression. The revised portion “Ready-to-eat rice (RER), which has a high porosity and low homogeneity, is frequently employed as a staple food in fast food. The investigation of the temperature distribution of RER during microwave reheating is helpful to provide a parameter basis for improving the microwave reheating effect of fast food.” has been marked in red font in the revised manuscript as shown in Line 60-64.
(3) In my opinion, the Introduction section is too superficial. Please discuss the role of dielectric parameters in temperature distribution and heating efficiency in the Introduction section, as well.
Response: Thanks for your valuable comment and suggestion. We add the disscusion “the role of dielectric parameters in temperature distribution and heating efficiency in the Introduction section” in Line 65-75.
The revised portion: “Food absorbs microwave energy and then transforms into heat energy during the microwave reheating process. Heat energy raises the temperature of food and facilitates water diffusion and evaporation. Food properties change affect the temperature distribution during the reheating process, and dielectric properties determine food's ability to absorb and transform microwave energy [22]. Studying the change of food dielectric properties is helpful to analyze the influence of microwave energy absorption on temperature distribution. In the process of microwave reheating, changes in thermal properties can indicate the law of heat consumption and transport. Food characteristic indexes are useful for clarifying the heat and mass transfer mechanism of food during microwave reheating, analyzing the causes of uneven temperature distribution, and providing a reasonable basis for improving uneven temperature distribution.”
(4) In line 90-91 check the parameters for calculation Eq.(3). (difference between the densities).
Response: We are very sorry for the error. The following is the correction “Porosity is defined as the ratio of pore volume in porous materials to the total volume in their natural state [27]. The porosity of RER was determined by Equation (1).
(1)
Where is the apparent molar volume of RER in its natural state (m3), V is the volume in the absolute density of RER (m3), represents the apparent density of RER (kg/m3) and represents the absolute density of RER (kg/m3).”, The revised portion in Equation (1) has been marked in red font.
- Please give the type of thermal characteristic analyser and condition of measurements (section 2.2.5).
Response: We are very sorry for our unclear expression. We add “the type of thermal characteristic analyser and condition of measurements” and marked in red font in the revised manuscript in Line 127, andLine 130-135.
(6) Please provide the details of dielectric measurements (analyser, conditions etc.).
Response: We are very sorry for our unclear expression. We add “the details of dielectric measurements (analyser, conditions etc.)” and marked in red font in the revised manuscript in Line 137, andLine 140-148.
[7.] In line 105 wavelength is mentioned, but is not used in Eq. (4).
Response: We are very sorry for the error. The following is the correction “Where is the free-space light velocity in m/s, represents the relative dielectric constant of RER and is relative dielectric loss of RER.” and marked in red font in the revised manuscript in Line 153-154.
(8) Please provide the details of microwave unit used for heating (frequency, positioning of packaged system etc.).
Response: Thanks for your valuable suggestion. We add “the details of microwave unit used for heating (frequency, positioning of packaged system etc.)” and marked in red font in the revised manuscript in Line 162-175.
(9) The dielectric constant and dielectric loss factor (and therefore the penetration depth) is also frequency dependent. Please provide the measuring frequency for these data (Figure 5,6,7).
Response: Thanks for your suggestion. We add the “the measuring frequency for these data (Figure 5,6,7)”. and marked in red font in the revised manuscript in Line 277-279 (Figure 5.).
(10) Establishments in section 3.5.1 need to be strengthened by references, as well.
Response: Thanks for your valuable suggestion. We add the “references”. and marked in red font in the revised manuscript in Line 352.
(11) Please add reference(s) to line 361-369.
Response: Thanks for your valuable suggestion. We add the “references”. and marked in red font in the revised manuscript in Line 418.
(12) Please check and correct the use of sub/superscript in the MS (see line 264, for instance).
Response: Thanks for your suggestion. We check and correct the use of sub/superscript in the Article, and marked in red font.
(13) The visibility of Figure 13-15 is very poor. Please improve it.
Response: We are sorry that “The visibility of Figure 13-15 is very poor”. We add the “Figure 14-16” and marked in red font in the revised manuscript in Line 385, 398, and 410.
- Results related to metal packaging structure design (section 3.6) need discussion with relevant references, as well.
Response: Thanks for your suggestion. We add the “discussion with relevant references” and marked in red font in the revised manuscript in Line 477, 486, and 488.
The other changes are marked in blue in the revised manuscript
We tried our best to improve the manuscript and made some other changes marked in blue in the revised manuscript. These changes will not depart from the main objectives and frameworks of the manuscript. We appreciate for Editors/Reviewers’ warm work earnestly, and hope that the revisions will meet with approval.
Once again, thank you very much for your comments and suggestions.
The other changes are marked in blue in the revised manuscript
We tried our best to improve the manuscript and made some other changes marked in blue in the revised manuscript. These changes will not depart from the main objectives and frameworks of the manuscript. We appreciate for Editors/Reviewers’ warm work earnestly, and hope that the revisions will meet with approval.
Once again, thank you very much for your comments and suggestions.

Round 2
Reviewer 2 Report
The authors decided all comments well. The revised manuscript can be considered for publication
Reviewer 3 Report
The manuscript has an interesting and relevant topic. Authors have revised the manuscript thoroughly according to reviewers' comments and suggestions and provided detailed answers for reviewers' questions. The revision made the manuscript more complete and clear. I agree and accept all modifications made by the authors.